# Pathological and Molecular Characteristics of Colorectal Cancer with Brain Metastases

**DOI:** 10.3390/cancers10120504

**Published:** 2018-12-10

**Authors:** Pauline Roussille, Gaelle Tachon, Claire Villalva, Serge Milin, Eric Frouin, Julie Godet, Antoine Berger, Sheik Emambux, Christos Petropoulos, Michel Wager, Lucie Karayan-Tapon, David Tougeron

**Affiliations:** 1Department of Radiation Oncology, University Hospital of Poitiers, 86021 Poitiers, France; pauline.roussille@chu-poitiers.fr (P.R.); antoine.berger@chu-poitiers.fr (A.B.); 2INSERM 1084, Experimental and Clinical Neurosciences Laboratory, University of Poitiers, 86073 Poitiers, France; gaelle.tachon@chu-poitiers.fr (G.T.); sheik.emambux@chu-poitiers.fr (S.E.); christospetropoulos81@hotmail.com (C.P.); michel.wager@chu-poitiers.fr (M.W.); lucie.karayan-tapon@chu-poitiers.fr (L.K.-T.); 3Faculty of Medicine, University of Poitiers, 86021 Poitiers, France; eric.frouin@chu-poitiers.fr; 4Cancer Biology Department, University Hospital of Poitiers, 86021 Poitiers, France; claire.villalva-gregoire@chu-poitiers.fr; 5Pathology Department, University Hospital of Poitiers, 86021 Poitiers, France; serge.milin@chu-poitiers.fr (S.M.); julie.godet@chu-poitiers.fr (J.G.); 6Medical Oncology Department, University Hospital of Poitiers, 86021 Poitiers, France; 7Department of Neurosurgery, University Hospital of Poitiers, 86021 Poitiers, France; 8Department of Gastroenterology, University Hospital of Poitiers, 86021 Poitiers, France

**Keywords:** brain metastases, colorectal cancer, *KRAS* mutation, PD-L1, tumor infiltrating lymphocytes

## Abstract

*Background:* Colorectal cancers (CRC) with brain metastases (BM) are scarcely described. The main objective of this study was to determine the molecular profile of CRC with BM. *Methods:* We included 82 CRC patients with BM. *KRAS*, *NRAS*, *BRAF* and mismatch repair (MMR) status were investigated on primary tumors (*n* = 82) and BM (*n* = 38). ALK, ROS1, cMET, HER-2, PD-1, PD-L1, CD3 and CD8 status were evaluated by immunohistochemistry, and when recommended, by fluorescence in situ hybridization. *Results:* In primary tumors, *KRAS*, *NRAS* and *BRAF* mutations were observed in 56%, 6%, and 6% of cases, respectively. No *ROS1*, *ALK* and *cMET* rearrangement was detected. Only one tumor presented *HER-2* amplification. Molecular profiles were mostly concordant between BM and paired primary tumors, except for 9% of discordances for *RAS* mutation. CD3, CD8, PD-1 and PD-L1 expressions presented some discordance between primary tumors and BM. In multivariate analysis, multiple BM, lung metastases and PD-L1+ tumor were predictive of poor overall survival. *Conclusions:* CRCs with BM are associated with high frequency of *RAS* mutations and significant discordance for *RAS* mutational status between BM and paired primary tumors. Multiple BM, lung metastases and PD-L1+ have been identified as prognostic factors and can guide therapeutic decisions for CRC patients with BM.

## 1. Introduction

Brain metastases (BM) from colorectal cancer (CRC) are rare with an incidence ranging from 0.6 to 3.2% and are associated with a poor prognosis with an overall survival (OS) of about 5.0 months [1,2]. Patients with BM from CRC present a specific clinical profile with predominant rectosigmoid primary tumor location and lung metastases [3,4,5,6]. Nevertheless, the molecular profile of BMs from CRC has only been partially explored [7,8]. Some small series have suggested a high rate of *KRAS* mutation in CRC with BM, but no study has evaluated complete *RAS* (*KRAS* and *NRAS*), *BRAF* and mismatch repair (MMR) status [1].

In metastatic CRC (mCRC), molecular profiles of liver and lung metastases have already been tested and revealed a high concordance between the metastases and paired primary tumor (PPT) (95–100%) [9]. Brastianos et al., by performing a whole-exome sequencing of 86 BM and PPT from various sites, reported 53% of discordances in genetic profile, and found actionable mutations (EGFR, HER-2 and PI3K/AKT/mTOR pathways) in BM that were not detected in PPT [10]. However, only four CRCs were analyzed. Therefore, it is of major interest to evaluate molecular abnormalities of CRC with BM in a larger cohort.

The main objective of this study was to evaluate the molecular profile of CRC with BM. The secondary objectives were to evaluate the concordance of molecular profiles between BM and their PPT and to determine the prognostic factors of CRC patients with BM.

## 2. Results

### 2.1. Patient and Tumor Characteristics

Eighty-two CRC patients with BM were included, mostly radiologically confirmed (*n* = 44/82), with a median follow-up of 45.1 months (95% Confidence Interval (CI) 26.6–45.5 months). Median age at CRC diagnosis was 64.0 years and most of the patients were male (63%) (Table 1).

### 2.2. Molecular and Pathological Profiles of Colorectal Cancer with Brain Metastases

In primary tumors (*n* = 82), RAS mutations were observed in 62% of cases with 56% of *KRAS* mutations and 6% of *NRAS* mutations (Table 2). *KRAS* mutations in codon 12 of exon 2 were observed in 48% and the most frequent were G12D and G12V. *BRAF* mutation was observed in 6%. Concerning BM (*n* = 38), RAS was mutated in 85% of cases (74% of *KRAS* mutations and 11% of *NRAS* mutations) and BRAF in 5%. Both primary tumors and BM were mostly MMR-proficient (pMMR) (95%). Four patients had dMMR tumors, one patient had a Lynch syndrome (*MSH2* germline mutation) and the three others patients had sporadic dMMR tumors.

No primary tumor overexpressed ROS1 protein according to immunohistochemistry (IHC) analysis. ALK IHC 1+ was detected in six primary tumors, but was negative by Fluorescence in situ hybridization (FISH) analysis. Concerning HER-2 IHC, three primary tumors were positive, but HER-2 amplification was confirmed by FISH only for one sample. cMET positive staining was detected by IHC in 61% of primary CRC, but none was confirmed by FISH. Concerning BM, ROS1, ALK and HER-2 staining were all negative (score 0). cMET positive staining was detected in 84% of BM, but none was confirmed by FISH.

Ten primary tumors (14%) were programmed death-1 positive (PD-1+), but no BM. Five primary tumors (7%) and two BMs (5%) were programmed death-ligand 1 positive (PD-L1+). Among the five PD-L1+ primary tumors, three were MMR-deficient (dMMR) and two were pMMR. The median percentage of CD3 and CD8 lymphocyte infiltrates were 30% and 11% in primary tumors, 11% and 3% in BM respectively. The mean percentages of CD3 and CD8 lymphocyte infiltrates in primary tumors were 46% and 38% in dMMR tumors and 33% and 11% in pMMR tumors (*p* = 0.23 for CD3 and *p* < 0.01 for CD8) respectively. The mean percentages of CD3 and CD8 lymphocyte infiltrates in primary tumors were 49% and 41% in PD-L1+ tumors and 33% and 12% in PD-L1- tumors (*p* = 0.09 for CD3 and *p* < 0.01 for CD8), respectively.

### 2.3. Concordance of Molecular and Pathological Profiles between Brain Metastases and Their Paired Primary Tumors

The molecular profiles of BM were compared with their PPT (Table 3), when available (*n* = 35). Discordances in *RAS* and *BRAF* status were observed in four patients (11%), three for *RAS* and one for *BRAF*. In each case, PPT was wild-type and BM was mutated. According to IHC evaluation, PPT and BM were discordant for cMET in nine cases (28%). However, all cases were negative according to FISH analyses.

Concerning PD-1+ tumor, discordance was observed in two paired samples (6%). We found three discordances for PD-L1 status (9%). Median percentages of CD3+ and CD8+ lymphocytes were significantly more important in PPT (34% and 10%) compared to BM (15% and 3%) (both *p* < 0.01). In addition, there was a positive correlation between levels of CD8+ infiltrates in BM and PTT (*p* = 0.01), but not for CD3+ infiltrates (*p* = 0.40).

### 2.4. Overall Survival

79 patients died at the time of data analysis. Median Overall Survival (OS) from BM diagnosis was 4.1 months (95%CI 3.6–5.4 months) (Figure 1). Median OS from diagnosis of metastatic disease was 28.6 months (95%CI 18.0–35.5 months). Age, BRAF mutation, PD-L1+ tumors, Eastern cooperative oncology group (ECOG) performance status ≥ 2, multiple BM and lung metastases were significantly associated with poor OS in univariate analysis (Table 4). In multivariate analysis, PD-L1+ primary tumors, multiple BM and lung metastases were significantly associated with poor OS.

## 3. Discussion

In our study, molecular features of CRC with BM were in accordance with rates observed in all-comers mCRC except for *RAS* mutations that appear to be higher than rates commonly observed in mCRC [11]. Surprisingly, we observed some differences of molecular profiles between BM and PPT, especially for *RAS* and PD-L1 status. Finally, we identified multiple BM, lung metastases and PD-L1 positivity as prognostic factors in patients with BM from CRC.

As compared to all-comers mCRC patients, in our study, patients with BM from CRC seemed to be younger, with more frequent rectal tumor and lung metastases. Other studies had previously identified frequent lung metastases and young age as particular characteristics of CRC patients with BM [1]. In accordance with the literature, the interval between primary tumor diagnosis and BM diagnosis reached more than 30 months, probably because the brain is a late sanctuary site for chemo-resistant tumor cells [12]. Moreover, the rate of *RAS* mutation was high (62%) in comparison to what is usually observed in mCRC (≈50%) [11]. This observation is in agreement with other studies, which also showed that *KRAS* mutations could be a predictive factor of BM [13]. The rates of CD3 and CD8 tumor-infiltrating lymphocytes (TILs) observed in our study were in accordance with the rates observed in other mCRC cohorts [14,15]. Also, our study showed comparable proportions of PD-L1+ and PD-1+ tumors, mostly in dMMR tumors, when compared with other studies in the literature [16,17].

There is a high discrepancy observed between IHC and FISH results for cMET status in our study, as described in the literature. In a recent study using IHC, 57.5% of CRC were found to be positive for MET protein IHC, but only 4.4% were FISH positive [18]. Overexpression of MET has been established in CRC [19], with MET protein levels ranging from 12% to 81% (median, 61%) [20]. Zeng et al. established that *MET* gene amplification was present in 2% of localized CRC tumors, 9% of tumors with distant metastases, and 18% of liver metastases using the quantitative PCR/ligase detection reaction technique [21]. In our study cMET positive staining was detected by IHC in 61% of primary CRC, but none was confirmed by FISH.

Comparison of BM and PPT has been scarcely explored in mCRC, but discordances have been observed between BM and PPT in lung and breast cancers [10]. In our study, we found a higher rate of *RAS* mutation in BM (85%) compared to PPT (62%) and three discordant cases (9%). El-Deiry et al. determined *KRAS* status from 2510 primary CRC and 30 BM from CRC and found significantly higher rates of *KRAS* mutation in BM (65%) compared to the primary tumor (45%), but the samples were not paired [13]. In another cohort of 41 BM with PPT, two cases presented discordant *KRAS* status [22]. Discordances between PPT and BM could be explained by intra and/or inter-tumoral heterogeneity, as we recently demonstrated in CRC [23]. Indeed, if CRC patients have had BM surgery, *RAS* should be evaluated in this sample in order to define treatment (anti-EGFR). BM are more frequently observed in breast and gastric cancers with HER-2 overexpression compared to HER-2 negative tumors [24,25], which does not seem to be the case in mCRC.

To our knowledge, no previous study has compared the expression of PD-1, PD-L1, CD3 and CD8 in paired primary CRC and BM. In BM, we identified low rates of immune infiltrates compared to PPT. These results were concordant with the study by Harter et al., which showed low rates of PD-L1+ and PD-1+ tumors (1%) and low rates of CD3+ (3%) and CD8+ T-cells (2%) in BM samples from CRC [26]. In the literature, whatever the tumor type, less immune infiltrate is observed in BM compared to PPT [27]. Moreover, in our study, there was some discordance between PD-1 and PD-L1 status in BM compared to PPT. Recent studies have identified BM as a sanctuary site for tumor cells to escape immunosurveillance [28]. Up until now, there has been only limited data concerning immune checkpoint inhibitor efficacy in BM, but no clinical evidence of lesser efficacy compared to other metastatic sites [29]. Nevertheless, it is important to consider the spatial heterogeneity of the tumor immune microenvironment in BM compared to PPT, especially PD-L1 expression, when cancer patients are treated with PD-1 or PD-L1 inhibitors.

Overall survival of patients with BM from CRC is poor. It is important to identify prognostic factors to help therapeutic decision-making. Some prognostic classifications exist, but most are not designed specifically for patients with mCRC. A recent Italian retrospective study identified age, performance status, BM site and BM number as prognostic factors associated with OS of CRC patients with BM [30]. In our study, we found no association between *RAS* or *BRAF* status and OS. However, OS of patients with PD-L1 negative primary tumors was significantly higher than patients with PD-L1+ tumors. This result should be interpreted with caution considering the small number of patients with PDL1+ tumors, the potential tumor heterogeneity and the absence of standard cut-off for this marker. High PD-L1 expression has been associated with longer OS in pMMR mCRC in some studies, but not all [31]. In addition, in lung cancer with BM, PD-L1 expression has been associated with worse OS [32]. Our study highlighted two other prognostic markers, single BM and the absence of lung metastases that had already been reported for patients with BM whatever the primary tumor.

The main limitation of the study is its retrospective nature, but there are few missing data (≈10%). Results concerning the comparison of BM and PPT should be confirmed given the small size of our study, since most patients did not have surgery of BM. Nevertheless, it is the largest study up until now concerning the molecular profile of CRC with BM.

## 4. Materials and Methods

### 4.1. Patients

All patients with BM from CRC, diagnosed from 2001 to 2016, were identified in our institution using our clinical report database. All patients with a histologically confirmed CRC and histologically or radiologically confirmed BM by computed tomography scan (CT-scan) and/or magnetic resonance imaging (MRI) were included. BM was defined as synchronous if they occur within three months of mCRC diagnosis. Our institution’s Ethics Committee approved the study (DC-2008-565).

### 4.2. Molecular Analyses

Genomic DNA from tumor samples was extracted using Maxwell 16 FFPE Plus LEV DNA purification kit^©^ (Promega, Charbonnières-les-Bains, France). *KRAS*/*NRAS* codons 12, 13, 61, 146 and *BRAF* (V600E) were analyzed by pyrosequencing (TheraScreenPyroKit^©^, Qiagen, Hilden, Germany) using homemade specific primers as previously described [33]. MMR status was determined by microsatellite analysis using MD1641 Promega kit^©^ (Promega).

### 4.3. Tissue Microarray Construction and Immunohistochemistry

Formalin-fixed paraffin-embedded blocks were used for tissue microarray (TMA) construction using four biopsy cores of 1 mm diameter per tumor in the tumor center (MTA Booster^©^ version 1.01, Alphelys, Paris, France).

IHCwas carried out on paraffin-embedded 3-μm thick TMA sections with antibodies directed against ALK, ROS1, cMet, HER-2, PD-1, PD-L1, CD3 and CD8 according to the manufacturer’s instructions.

IHC is a prescreening test commonly used for the detection of ALK rearrangement in lung carcinoma [34] and the same scoring was used here. Immunostaining scores were assigned from 0 to 3. For ALK cytoplasmic staining, a score of 1+ (weak), 2+ (moderate) or 3+ (strong) in more than 10% of tumor cells and for ROS1 staining, any percentage of tumor cells with cytoplasmic staining intensity of 1+, 2+ or 3+ were considered as IHC-positive and then evaluated by FISH [35]. Indeed, FISH is considered the “gold standard” to confirm IHC results, due to possible false-positive signals with IHC testing [36]. For MET only 2+ or 3+ in more than 10% of tumor cells were defined as positive and subsequently evaluated by FISH [37]. HER-2 IHC positive status was defined as tumors with a 2+ or 3+ staining in more than 10% of the cells and then evaluated by FISH [38].

PD-1 IHC was considered positive when ≥1% of intra-epithelial TILs were stained. PD-L1 immunostaining was considered positive when ≥1% of tumor cells had membranous staining [39]. CD3 and CD8 staining were also analyzed as the percentage of both intra-tumoral and stromal CD3 and CD8 positive lymphocytes over the total immune cells [14,15].

### 4.4. Fluorescent In Situ Hybridization (FISH)

Vysis ALK Break Apart FISH probes^©^ (Abbott Molecular, Abbott Park, IL, USA), HER-2/CEP17 DNA Probe Kit II probes^©^ (Abbott Molecular) and ZytoLight SPEC MET/CEN 7 Dual Color Probes^©^ (ZytoVision, Bremerhaven, Germany) were used respectively for the detection of *ALK* rearrangement, *HER-2* and *cMET* amplification.

*ALK* locus rearrangement was considered translocated if ≥15% of tumor cells showed isolated red signal(s) and/or split red and green signals. *ALK* appeared amplified and required further verification if an average copy number ≥6 copies per nucleus was detected [40]. HER-2 was considered amplified if average *HER-2/CEP17* ratio was higher than 2.0 [38]. Tumors with *MET*/CEP7 ratio ≥2 or with an average number of *MET* signals per nucleus >6 were scored as positive for *MET* amplification [41].

### 4.5. Statistical Analysis

Survival curves and 95% confidence intervals were determined using the Kaplan-Meier method. Predictive factors of OS were evaluated using the log-rank test for univariate analysis and statistically significant variables were included in multivariate analysis using a Cox regression model. The level of significance was set at a *p* value of 0.05. Statistical analyses were performed using XLSTAT 2017 software (Addinsoft, New York, NY, USA).

## 5. Conclusions

Our study provided relevant and specific features of CRC patients with BM, such as frequent lung metastasis, frequent rectal tumor site and high rate of *RAS* mutation. These results suggest a need for BM screening in this mCRC patients subgroup, but will require further prospective investigations to determine if early identification of BM improves survival and/or quality of life. We have highlighted the usefulness of BM number, the presence of lung metastases and the expression of PD-L1 as prognostic markers. For the first time, we found that PD-L1 expression was associated with poor prognostic in CRC patients with BM. All of these new data can guide therapeutic decision-making in patients with BM from CRC.

## Figures and Tables

**Figure 1 cancers-10-00504-f001:**
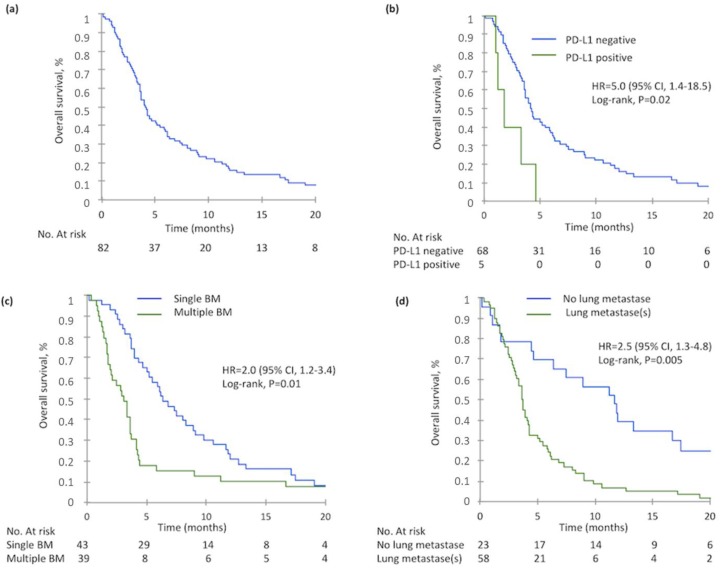
Overall Survival at brain metastasis(es) diagnosis in the whole population and according to PD-L1 expression, number of brain metastasis(es) and the presence of lung metastasis(es): (**a**) Overall survival of 82 patients at BM diagnosis, (**b**) Overall survival according to PD-L1 expression in primary tumor, (**c**) Overall survival according to the BM number, (**d**) Overall survival according to the presence of lung metastasis(es) at BM diagnosis.

**Table 1 cancers-10-00504-t001:** Clinical characteristics of patients, primary tumors and brain metastases (BM).

Characteristics	Patients (*n* = 82)
**Age at primary tumor diagnostic, years**	
Median (range)	64 (35–85)
**Gender, *n* (%)**	
Male	52 (63)
Female	30 (37)
**Site of primary tumor, *n* (%)**	
Ascending colon	19 (23)
Descending colon	24 (29)
Rectum	35 (42)
Bifocal tumor	5 (6)
**Tumor grade, *n* (%)**	
Well or moderately differentiated	61 (87)
Poorly differentiated	9 (13)
Missing	12
**Stage at initial CRC diagnostic, *n* (%)**	
I	4 (5)
II	13 (16)
III	26 (32)
IV	39 (47)
**Primary tumor resection, *n *(%)**	
No	11 (13)
Yes	71 (87)
**ECOG performance status at BM diagnosis, *n* (%)**	
< 2	43 (54)
≥ 2	36 (46)
Missing	3
**Number of BM, *n* (%)**	
Single	43 (52)
Multiple	39 (48)
**Site of BM, *n* (%)**	
Supratentorial	46 (56)
Subtentorial	18 (22)
Both	18 (22)
**Delay between BM and CRC diagnosis, *n* (%)**	
Synchronous	8 (10)
Metachronous	74 (90)
**ECM at BM diagnosis, *n* (%)**	
No	11 (14)
Yes	70 (86)
Missing	1
**Lung metastases at BM diagnosis, *n* (%)**	
No	23 (28)
Yes	58 (72)
Missing	1
**Liver metastases at BM diagnosis, *n* (%)**	
No	45 (56)
Yes	36 (44)

Abbreviations: BM, brain metastasis(es); CRC, colorectal cancer; ECM, extracranial metastasis(es); ECOG, Eastern Cooperative Oncology Group score.

**Table 2 cancers-10-00504-t002:** Molecular profile of primary tumors and brain metastases.

Molecular Status	Primary Tumors (*n* = 82)	BM (*n* = 38)
***KRAS* status**		
Wild-type, *n* (%)	35 (44)	10 (26)
Mutant, *n* (%)	44 (56)	28 (74)
*KRAS* exon 2 at codon 12		
G12D	14 (18)	9 (23)
G12V	14 (18)	8 (21)
G12A	5 (6)	3 (8)
G12S	3 (4)	0
G12C	1 (1)	1 (3)
G12R	1 (1)	1 (3)
*KRAS* exon 2 at codon 13		
G13D	2 (3)	3 (8)
G13R	1 (1)	1 (3)
*KRAS* exon 3 at codon 61	3 (4)	2 (5)
*KRAS* exon 4 at codon 146	0	0
Missing, *n*	3	0
***NRAS* status**		
Wild-type, *n* (%)	74 (94)	34 (89)
Mutant, *n* (%)	5 (6)	4 (11)
*NRAS* exon 2 at codon 12 or 13	1 (1)	1 (3)
*NRAS* exon 3 at codon 61	4 (5)	3 (8)
Missing, *n*	3	0
***BRAF* exon 15 at codon 600**		
Wild-type, *n* (%)	74 (94)	36 (95)
Mutant, *n* (%)	5 (6)	2 (5)
Missing, *n* (%)	3	0
**MMR status**		
pMMR, *n* (%)	70 (95)	36 (95)
dMMR, *n* (%)	4 (5)	2 (5)
Missing, *n*	8	0
**cMET expression**		
Negative (0, 1+, 2+/3+ with FISH negative), *n* (%)	76 (100)	37 (100)
Positive (2+, 3+ with FISH positive), *n* (%)	0	0
Missing, *n*	6	1
**HER-2 expression**		
Negative (0, 1+, 2+ with FISH negative), *n* (%)	74 (99)	37 (100)
Positive (2+ with FISH positive, 3+), *n* (%)	1 (1)	0
Missing, *n*	7	1
**ALK expression**		
Negative (0, 1+/2+/3+ with FISH negative), *n* (%)	76 (100)	37 (100)
Positive (1+/2+/3+ with FISH positive), *n* (%)	0	0
Missing, *n* (%)	6	1
**ROS1 expression**		
Negative (0, 1+/2+/3+ with FISH negative), *n* (%)	74 (100)	37 (100)
Positive (1+/2+/3+ with FISH positive), *n* (%)	0	0
Missing, *n*	8	1
**PD-1 expression**		
Negative, *n* (%)	64 (86)	37 (100)
Positive, *n* (%)	10 (14)	0
Missing, *n*	8	1
**PD-L1 expression**		
Negative, *n* (%)	68 (93)	35 (95)
Positive, *n* (%)	5 (7)	2 (5)
Missing, *n*	9	1
**CD3 expression**		
Median rate, % (range)	30 (0–80)	11 (0–60)
Missing, *n*	11	1
**CD8 expression**		
Median rate, % (range)	11 (0–70)	3 (0–50)
Missing, *n*	7	2

Abbreviations: IHC, Immunohistochemistry; FISH, Fluorescence in situ hybridization; MMR, Mismatch repair; pMMR, Proficient Mismatch Repair; dMMR, Deficient Mismatch Repair.

**Table 3 cancers-10-00504-t003:** Molecular and pathological profiles of brain metastases and paired primary tumors.

	Brain Metastases		
***RAS* status**			
**Primary tumors**	Wild-type	Mutant	Total
Wild-type, *n* (%)	6 (17)	3 (9)	9 (26)
Mutant, *n* (%)	0	26 (74)	26 (74)
Total, *n* (%)	6 (17)	29 (83)	35
***BRAF* status**			
**Primary tumors**	Wild-type	Mutant	Total
Wild-type, *n* (%)	33 (94)	1 (3)	34 (97)
Mutant, *n* (%)	0	1 (3)	1 (3)
Total, *n* (%)	33 (94)	2 (6)	35
**MMR status**			
**Primary tumors**	pMMR	dMMR	Total
pMMR, *n* (%)	30 (94)	0	30 (94)
dMMR, *n* (%)	0	2 (6)	2 (6)
Total, *n* (%)	30 (94)	2 (6)	32
**HER-2 expression**			
**Primary tumors**	Negative	Positive	Total
Negative, *n* (%)	35 (100)	0	35 (100)
Positive, *n* (%)	0	0	0 (0)
Total, *n* (%)	35 (100)	0 (0)	35
**cMET expression (IHC)**			
**Primary tumors**	Negative	Positive	Total
Negative, *n* (%)	4 (13)	7 (22)	11 (34)
Positive, *n* (%)	2 (6)	19 (59)	21 (66)
Total, *n* (%)	6 (19)	26 (81)	32
**PD-1 expression**			
**Primary tumors**	Negative	Positive	Total
Negative, *n* (%)	30 (94)	0	30 (94)
Positive, *n* (%)	2 (6)	0	2 (6)
Total, *n* (%)	32 (100)	0	32
**PD-L1 expression**			
**Primary tumors**	Negative	Positive	Total
Negative, *n* (%)	29 (91)	2 (6)	31 (97)
Positive, *n* (%)	1 (3)	0	1 (3)
Total, *n* (%)	30 (94)	2 (6)	32
**CD3 expression**	**Primary tumor**	**Brain metastases**	
Median rate, % (range)	34 (0–80)	15 (0–60)	
**CD8 expression**	**Primary tumor**	**Brain metastases**	
Median rate, % (range)	10 (0–70)	3 (0–50)	

Abbreviations: IHC, immunohistochemistry; MMR, Mismatch repair; pMMR, Proficient Mismatch Repair; dMMR, Deficient Mismatch Repair.

**Table 4 cancers-10-00504-t004:** Univariate and multivariate analysis of overall survival in patients with brain metastases from colorectal cancer.

		Univariate Analysis		Multivariate Analysis		
Variables	*n*	Median (Months)	*p* Value	HR	95% CI	*p* Value
**Gender (n = 82)**			0.79 *			0.38
Male	52	3.9		1		
Female	30	4.3		0.8	0.5–1.4	
**Age at BM diagnosis (n = 82)**	82		0.02 *	1.0	1.0–1.0	0.62
**Site of primary tumor (n = 82)**			0.23			
Ascending colon	20	4.5				
Descending colon	24	5.9				
Rectum	35	2.9				
**Tumor grade (n = 70)**			0.05			
Well or moderately differentiated	61	3.9				
Poorly differentiated	9	4.6				
***RAS* status (n = 79)**			0.65			
Wild-type	30	3.6				
Mutant	49	4.3				
***BRAF* status (n = 79)**			0.03 *			0.76
Wild-type	74	4.2		1		
Mutant	5	3.3		1.2	0.3–4.2	
**MMR status (n = 74)**			0.68			
pMMR	70	4.1				
dMMR	4	4.0				
**PD-1 expression (n = 74)**			0.79			
Negative	64	4.2				
Positive	10	3.6				
**PD-L1 expression (n = 73)**			0.009 *			0.02
Negative	68	4.2		1		
Positive	5	1.8		5.0	1.4–18.5	
**CD3 expression (n = 71)**	71		0.08			
**CD8 expression (n = 75)**	75		0.45			
**ECOG performance status (n = 79)**			0.0003 *			0.07
<2	43	7.3		1		
≥2	36	3.2		1.8	1.0–3.4	
**Number of BM (n = 82)**			0.003 *			0.01
Single	43	6.3		1		
Multiple	39	3.1		2.0	1.2–3.4	
**Lung metastases at BM diagnosis (n = 81)**			0.0003 *			0.005
No	23	11.7		1		
Yes	58	3.6		2.5	1.3–4.8	
**Liver metastases at BM diagnosis (n = 81)**						
No	45	4.3	0.31			
Yes	36	3.7				

Abbreviations: HR, hazard ratio; BM, brain metastasis(es); 95% CI, 95% confidence interval; ECOG, Eastern Cooperative Oncology Group score. * variables included in multivariate analysis

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
