# Peer review of "Pathological and Molecular Characteristics of Colorectal Cancer with Brain Metastases"

_cancers, 2018, doi:10.3390/cancers10120504_

Reviewer 1 Report

Colorectal cancer (CRC) is the 3rd most incident malignant neoplasia worldwide, however our knowledge about the molecular pathways involved in CRC brain metastasis are still very scarce. In the study submitted by Pauline Roussille and collegues the author's main focus was to evaluate the molecular profile concordance between CRC brain metastases and their primary tumors (PT). For that the authors performed a retrospective study using the clinical information from approximately 82 patients.

Although aware of the restricted cohort size, the authors were still able to establish a relevant detailed molecular profile of mutations between matched PTs and theirs brain metastases. Molecular profiles were mostly concordant between PTs and their brain metastasis, with exception of approximately 9% discordance for RAS mutation.

This study can be considered as a base for further discussions on the need of screening BM from CRC patients, as the metastases of these patients might fall in another molecular signature category.

In general the study shows a clear aim and has executed its methodology correctly. I have minor suggestions for the authors:

1.  In the abstract, p.1, line 30 its written: “… , and when recommended, by fluorescence in situ hybridization”

- Please, briefly explain in the main text/results in which situations and why it is recommended to use FISH in addition to IHC

2.  Being aware of:

- Prasanna T., et al. Acta Oncol. 2018 Jul 23:1-
-Jianhua L., et al. Gastroenterology Research and Practice. Volume 2018, article ID 4585802

- Brastianos P.K., et al. Cancer Discor 2015, 5, 1164-1177

I strong suggest the authors to rephrase line 49, p2 “…has never been clearly exposed”.

3.  It would be relevant to further or better explain the discrepancies observed between IHC and FISH results, as this situation is mentioned several times in the results section.

4.  Last, in the discussion it is not clear what the authors want to state in the last paragraph of p9, lines 139-141. Please clarify/rewrite this.

Author Response

Reviewer  1:

Comments and Suggestions for Authors

Colorectal cancer (CRC) is the 3rd most incident malignant neoplasia worldwide, however our knowledge about the molecular pathways involved in CRC brain metastasis are still very scarce. In the study submitted by Pauline Roussille and collegues the author's main focus was to evaluate the molecular profile concordance between CRC brain metastases and their primary tumors (PT). For that the authors performed a retrospective study using the clinical information from approximately 82 patients.

Although aware of the restricted cohort size, the authors were still able to establish a relevant detailed molecular profile of mutations between matched PTs and theirs brain metastases. Molecular profiles were mostly concordant between PTs and their brain metastasis, with exception of approximately 9% discordance for RAS mutation.

This study can be considered as a base for further discussions on the need of screening BM from CRC patients, as the metastases of these patients might fall in another molecular signature category.

In general the study shows a clear aim and has executed its methodology correctly. I have minor suggestions for the authors:

1. In the abstract, p.1, line 30 its written: “… , and when recommended, by fluorescence in situ hybridization”.

-  Please, briefly explain in the main text/results in which situations and why it is recommended to use FISH in addition to IHC  

This information was already partly explained in paragraph "4.3. Tissue microarray construction and immunohistochemistry". For lung adenocarcinoma, it is commonly admitted to screen for ALK rearrangement with immunohistochemistry (IHC) method followed by FISH confirmation due to possible false-positive signals with IHC testing. The same method was used in our study. This information has been added in the manuscript Page 10. Line 217-223.

2.  Being aware of:

- Prasanna T., et al. Acta Oncol. 2018 Jul 23:1-7 Added as ref 7. Page 2. Line 49.

- Jianhua L., et al. Gastroenterology Research and Practice. Volume 2018, article ID 4585802 Added as ref 8. Page 2. Line 49.[dt1] 

- Brastianos P.K., et al. Cancer Discor 2015, 5, 1164-1177 Already cited as ref 10. Page 2. Line 56.

I strong suggest the authors to rephrase line 49, p2 “…has never been clearly exposed”.   These 2 original articles are now included in the reference and the sentence has been modified "Nevertheless, the molecular profile of BMs from CRC has only been partially explored."  Page 2. Line 48.

3. It would be relevant to further or better explain the discrepancies observed between IHC and FISH results, as this situation is mentioned several times in the results section. 

These discrepancies concern the cMET status. As already known there is significant discrepancies observed between IHC and FISH results for cMET status. For example, in a recent study using IHC, 57.5% of CRC were found to be positive for MET protein IHC but only 4.4% were FISH positive (1). Overexpression of MET has been established in CRC (2), with MET protein levels ranging from 12% to 81% (median, 61%) (3). Zeng et al. established that MET gene amplification was present in 2% of localized CRC tumors, 9% of tumors with distant metastases, and 18% of liver metastases using the quantitative PCR/ligase detection reaction technique (4). The discrepancies observed for cMET status between IHC and FISH results is now discussed in the manuscript. Page 9. Line 148-155.

(1) [ Zhang, M,Li, G,Sun, X, et al. MET amplification, expression, and exon 14 mutations in colorectal adenocarcinoma. Hum Pathol 2018;77:108–115.]

(2) [Abou-Bakr AA, Elbasmi A. c-MET overexpression as a prognostic bio- marker in colorectal adenocarcinoma. Gulf J Oncolog 2013;1:28-34.]

(3) [Liu Y, Yu XF, Zou J, Luo ZH. Prognostic value of c-Met in colorectal cancer: a meta-analysis. World J Gastroenterol 2015;21:3706-10.]

(4) [Zeng ZS, Weiser MR, Kuntz E, et al. c-Met gene amplification is associated with advanced stage colorectal cancer and liver metastases. Cancer Lett 2008;265:258-69.]

4. Last, in the discussion it is not clear what the authors want to state in the last paragraph of p9, lines 139-141. Please clarify/rewrite this.  

The sentence “The rates of CD3 and CD8 TIL observed in our study were in accordance with mCRC data [14,15]. In the literature, PD-L1+ tumors represent 10%-20% of CRC and PD-1+ tumors 20%-50% of CRC, with higher proportions in dMMR CRC [16,17]. Our study showed comparable proportions of PD-L1+ and PD-1+ tumors, mostly in dMMR tumors.” has been modified to “The rates of CD3 and CD8 tumor-infiltrating lymphocytes (TILs) observed in our study were in accordance with the rates observed in other mCRC cohorts [14,15]. Also, our study showed comparable proportions of PD-L1+ and PD-1+ tumors, mostly in dMMR tumors, when compared with other studies in the literature [16,17].”

Reviewer 2 Report

The paper by Roussille et al. reports a study on prognostic factors of colorectal cancer with brain metastases.

The authors used several clinical, pathological, and molecular determinants of prognosis.

Mutations in some genes involved in CRC and BM are examined, in order to evaluate concordance between PT and metastases. Then they evaluate the weight of each variable in determine OS in univariate and multivariate analyses. The work is concise but well balanced. Several approaches are examined both clinical and molecular. The number of tumors allows to draw sound conclusions, though breakdown analyses reduce subgroup numbers. The work adds a piece of knowledge to the overall picture of CRC with brain metastases. The overall rate of the study is good. The main drawback is its retrospective design.

I have minor suggestions and questions:

1.       Did patients with dMMR concordant tumors have PMS2 mutations? Were they younger than the others? In other words, were Lynch and/or Turcot syndromes excluded?

2.       A few minor spelling errors should be corrected, especially in the discussion section of the paper.

Author Response

Comments and Suggestions for Authors

The paper by Roussille et al. reports a study on prognostic factors of colorectal cancer with brain metastases.

The authors used several clinical, pathological, and molecular determinants of prognosis.

Mutations in some genes involved in CRC and BM are examined, in order to evaluate concordance between PT and metastases. Then they evaluate the weight of each variable in determine OS in univariate and multivariate analyses. The work is concise but well balanced. Several approaches are examined both clinical and molecular. The number of tumors allows to draw sound conclusions, though breakdown analyses reduce subgroup numbers. The work adds a piece of knowledge to the overall picture of CRC with brain metastases. The overall rate of the study is good. The main drawback is its retrospective design.

I have minor suggestions and questions:

1.       Did patients with dMMR concordant tumors have PMS2 mutations? Were they younger than the others? In other words, were Lynch and/or Turcot syndromes excluded?

In our study, 4 patients had a dMMR tumors. One patient had a Lynch syndrome (MSH2 germline mutation) aged 41 years old at the time of diagnosis. The three others patients aged 72, 73 and 81 years old respectively were sporadic dMMR tumors with BRAF mutation. This information has been added in the manuscript Page 3. Line 76-77.

2.       A fewminor spelling errors should be corrected, especially in the discussion section of the paper.  

The manuscript has been checked and detected spelling errors have been corrected.